# Comparison of Placental HSD17B1 Expression and Its Regulation in Various Mammalian Species

**DOI:** 10.3390/ani13040622

**Published:** 2023-02-10

**Authors:** Takashi Yazawa, Mohammad Sayful Islam, Yoshitaka Imamichi, Hiroyuki Watanabe, Kazuhide Yaegashi, Takanori Ida, Takahiro Sato, Takeshi Kitano, Shigenori Matsuzaki, Akihiro Umezawa, Yuki Muranishi

**Affiliations:** 1Department of Biochemistry, Asahikawa Medical University, Asahikawa 078-8510, Hokkaido, Japan; 2Department of Marine Bioscience, Fukui Prefectural University, Obama 917-0003, Fukui, Japan; 3Department of Life and Food Science, Obihiro University of Agriculture and Veterinary Medicine, Obihiro 080-8555, Hokkaido, Japan; 4Kamui Dental Clinic, Asahikawa 070-8012, Hokkaido, Japan; 5Center for Animal Disease Control, Frontiers Science Research Center, University of Miyazaki, Miyazaki 889-1692, Miyazaki, Japan; 6Division of Molecular Genetics, Institute of Life Sciences, Kurume University, Kurume 830-0011, Fukuoka, Japan; 7Department of Biological Sciences, Graduate School of Science and Technology, Kumamoto University, Kumamoto 860-8555, Kumamoto, Japan; 8Institute of Tokachi Breeding Techniques, Obihiro 080-0838, Hokkaido, Japan; 9Department of Reproduction, National Center for Child Health and Development Research Institute, Setagaya 157-8535, Tokyo, Japan

**Keywords:** HSD17B1, placenta, human, ovine, androstenedione, testosterone

## Abstract

**Simple Summary:**

17β-Hydroxysteroid dehydrogenase type1 (HSD17B1) is responsible for converting estrone, a weak estrogen, into the more potent estradiol in the ovaries of vertebrates ranging from teleosts to mammals. We evaluated its expression in the placentae of various eutherian species and demonstrated that it was only expressed in the human placenta. The ubiquitous transcription factor SP1-binding site was conserved in the promoter region of the *HSD17B1*/*Hsd17b1* gene across the species. This region conferred the promoter activities both in human and ovine *HSD17B1*. However, in the ovine placenta, cytosine residues of this SP1 site in the *HSD17B1* promoter region were completely methylated. Human HSD17B1 had high activity for testosterone production, similar to its homologs in other species. On the basis of this fact, the significance of placental *HSD17B1* expression is discussed.

**Abstract:**

During mammalian gestation, large amounts of progesterone are produced by the placenta and circulate for the maintenance of pregnancy. In contrast, primary plasma estrogens are different between species. To account for this difference, we compared the expression of ovarian and placental steroidogenic genes in various mammalian species (mouse, guinea pig, porcine, ovine, bovine, and human). Consistent with the ability to synthesize progesterone, CYP11A1/Cyp11a1, and bi-functional HSD3B/Hsd3b genes were expressed in all species. CYP17A1/Cyp17a1 was expressed in the placenta of all species, excluding humans. CYP19A/Cyp19a1 was expressed in all placental estrogen-producing species, whereas estradiol-producing HSD17B1 was only strongly expressed in the human placenta. The promoter region of *HSD17B1* in various species possesses a well-conserved SP1 site that was activated in human placental cell line JEG-3 cells. However, DNA methylation analyses in the ovine placenta showed that the SP1-site in the promoter region of *HSD17B1* was completely methylated. These results indicate that epigenetic regulation of HSD17B1 expression is important for species-specific placental sex steroid production. Because human HSD17B1 showed strong activity for the conversion of androstenedione into testosterone, similar to HSD17B1/Hsd17b1 in other species, we also discuss the biological significance of human placental HSD17B1 based on the symptoms of aromatase-deficient patients.

## 1. Introduction

Steroid hormones are produced from cholesterol in a series of steps catalyzed by cytochrome P450 hydroxylases and hydroxysteroid dehydrogenases (Figure 1). Among these steroidogenic genes, 17β-hydroxysteroid dehydrogenases (HSD17B), which include at least 14 family members, have been identified as enzymes that interconvert 17-keto and 17-hydroxy sex steroids [1]. Of these family members, human HSD17B1 is mainly expressed in ovarian granulosa cells and the placenta to regulate the production of active sex steroids [2]. HSD17B1 efficiently catalyzes the conversion of the weak estrogen, estrone (E1), into active estradiol (E2). In addition, it converts androstenedione (A4) to testosterone (T). The rodent Hsd17b1 enzyme catalyzes the conversion of A4 into T at a similar efficiency to that of E1 to E2, whereas it was indicated that the catalytic efficiency of human HSD17B1 for the A4 to T reaction is very weak or barely detectable [3,4], although contrasting results have recently been reported [5].

The placenta produces various hormones that play important roles in the maintenance of pregnancy [6]. Among them, progesterone is one of the steroid hormones which is commonly essential in all eutherian species during pregnancy [7,8]. It is produced from cholesterol through conversion from pregnenolone in a series of steps by cytochrome P450 family 11 subfamily A member 1 (CYP11A1/Cyp11a1) and 3β-hydroxysteroid dehydrogenases (HSD3B/Hsd3b). Progesterone inhibits uterine contraction, thereby preventing preterm labor. In addition, the placenta produces another steroid hormone, estrogen, in a species-specific fashion [8]. With the aid of HSD17Bs, the estrogens are produced by aromatase (CYP19A/Cyp19a1), following the conversion of pregnenolone and progesterone into androgens (dehydroepiandrosterone and A4) by CYP17A1/Cyp17a1. Although plasma estrogen levels are markedly increased at late pregnancy for parturition, their profiles are different between species. Estrone (E1) and its sulfate form are the most abundant estrogens in almost animal species [9,10,11,12], whereas estradiol (E2) is predominant in humans [13] and some rodents, such as mice and rats [14,15]. In these rodents, this is caused by the unique phenomenon that ovaries continue to represent the main source of plasma estrogens during pregnancy because of the deficiency of placental Cyp19a1 expression [16,17]. Since HSD17B1/Hsd17b1/hsd17b1 is responsible for the ovarian E2 production in vertebrates ranging from teleosts to mammals [2,18,19,20], there is a possibility that placental expression of this gene may account for the differences in major plasma estrogens between eutherian species during pregnancy. In this study, we compared the ovarian and placental expression of steroidogenic genes, including *HSD17B1*, in humans and other species. We also compared the transcriptional regulation of human and ovine *HSD17B1* in the placenta. In addition, we discussed the biological significance of human placental HSD17B1 based on the evaluation of its enzymatic activities and symptoms of aromatase-deficient patients.

## 2. Materials and Methods

### 2.1. Animals and Tissue Collection

This study was conducted in accordance with the National Institutes of Health Guide for Care and Use of Laboratory Animals following protocols approved by the Obihiro University of Agriculture and Veterinary Medicine Committee on Animal Care (permission number: 20–25, 22–48) and the Animal Experiment Committee of Asahikawa Medical University (J22-009). Mouse ovarian tissues were collected from female C57/BL6 mice (5–10 weeks old, *n* = 5) that are provided by Animal Laboratory for Medical Research at Asahikawa Medical University. Ovine fresh placenta just after parturition (*n* = 2) and ovarian tissues (12 months and 5 months old, *n* = 2) were collected from National Livestock Breeding Center and the farms in Obihiro (Hokkaido, Japan) and at a local abattoir (Asahikawa, Hokkaido, Japan). Porcine (*n* = 2) and bovine (*n* = 2) fresh placenta just after parturition were collected from the slaughterhouse in Obihiro. Porcine (6 months old, *n* = 3) and bovine (28–30 months old, *n* = 3) ovaries were provided by the Takasaki Meat Inspection Center (Miyazaki, Japan). Ovaries and placentae of guinea pigs (8 weeks old and 20–25 dpc, *n* = 2) were purchased from Sankyo Labo Service Corporation (Tokyo, Japan).

### 2.2. Reverse Transcriptase-Polymerase Chain Reaction (RT-PCR)

The total RNA of each whole tissue was extracted using RNA extracting reagents, TRIzol reagent (Thermo Fisher Scientific, Waltham, MA, USA), and TRIsure reagent (Bioline, Luckenwalde, Germany). Total RNA of the human ovary (20–60 years old, *n* = 17) and placenta (*n* = 15) were purchased from Takara Bio Inc. (Shiga, Japan) and BioChain Institute Inc. (Hayward, CA, USA). Total RNA of the porcine placenta (*n* = 1) and bovine placenta (*n* = 1) just after parturition were purchased from Zyagen (San Diego, CA, USA). The total RNA of the mouse placenta was purchased from Takara Bio Inc. and Zyagen (13–15 dpc, *n* = 3). RT-PCR was performed at least in triplicate in experiments [21,22]. SuperScript III Reverse Transcriptase (Thermo Fisher Scientific) was used for the cDNA synthesis from the total RNA of each tissue. PCR amplification of each gene from synthesized cDNA (25 ng/reaction) as a template was performed using gene-specific primers and Ex Taq (Takara Bio Inc.). The reaction products were separated by electrophoresis using a 1.25% agar gel, and the resulting bands were visualized by staining with ethidium bromide. The primers used for PCR are described in Appendix A. Other primers were described previously [21,22].

### 2.3. Cell Culture, Transfection, and Luciferase Assay

JEG-3, HEK293, and CV-1 cells were cultured in Dulbecco’s Modified Eagle Medium (DMEM, Nacalai Teaque Inc., Kyoto, Japan) supplemented with 10% fetal bovine serum (FBS, Nichirei Bioscience Inc., Tokyo, Japan) and antibiotics (Penicillin-Streptomycin, Nacalai Teaque Inc.). Transfection of these cells was performed by lipofection using HilyMax (Dojindo Laboratories, Kumamoto, Japan). One day before transfection, cells were seeded on 48-well plates and cultured in DMEM supplemented with 10% normal FBS or Hyclone Charcoal/Dextran-treated FBS (GE Healthcare U.K. Ltd., Buckinghamshire, England). In the experiments for evaluating the enzymatic activities of HSD17B1/Hsd17b1, we adopted a method that quantifies the conversion of A4 into T via AR-mediated transactivation [5]. Green fluorescent protein (GFP) or HSD17B1/Hsd17b1 expression vectors were transfected into HEK293 cells. After 2 days post-transfection, HEK293 cells were treated with A4 at 1 nM for 2 h. CV-1 cells were transfected with androgen response element (ARE)-Luc and human androgen receptor (AR)- expression vectors. After 24 h post-transfection, CV-1 cells were incubated for 24 h with vehicle (EtOH), A4 (1 nM, Sigma-Aldrich, St. Louis, MO, USA), T (1 nM, Sigma-Aldrich), culture medium derived from GFP or each HSD17B1/Hsd17b1-expressing HEK293 cells collected at 2 h after A4 addition. JEG-3 cells were harvested 48 h after transfection. Luciferase assays were performed as described previously [21]. Luciferase activity was determined by a dual luciferase reporter assay system (Progema Corporation, Carlsbad, CA, USA) using a MiniLumat LB9506 (Berthold Systems, Aliquippa, PA, USA) in a single tube, with the first assay of firefly luciferase, followed by the *Renilla* luciferase assay. Firefly luciferase activities (relative light units) were normalized by *Renilla* luciferase activities. Each data point represents the mean of at least four independent experiments.

### 2.4. Methylation Analyses

To convert all the cytosines except for methylated cytosines at the CpG dinucleotides into thymine following uracils, ovine placenta-derived genomic DNA samples were treated with the Epitect Fast Bisulfite Kit (Qiagen, Hilden, Germany). The 5′-upstream region of the *HSD17B1* gene surrounding the SP1 site was amplified using Takara Ex Premier DNA polymerase (Takara Bio Inc.) with the specific primers (Appendix A). Then, deoxyadenosines were added to each *3*′- end, and they were subcloned into the pGEM-T easy vector (Promega Corporation). Multiple clones were subjected to direct sequencing with the BigDye Terminator v3.1 Cycle Sequencing Kits (Applied Biosystems, Waltham, MA, USA) on the ABI3500 Genetic Analyzer (Applied Biosystems).

### 2.5. Alignment Analysis and Plasmids

The alignment analysis of the *HSD17B1* gene and its 5′-flanking region was performed using Clustal W. Accession numbers of analyzed nucleotides are as follows: human *HSD17B1* gene (M27138.1), mouse *Hsd17b1* gene (AF363242.1), porcine chromosome 12 including *HSD17B1* gene (NC_010454.4), bovine chromosome 19 including *HSD17B1* gene (LR962874.1), and ovine chromosome 11 including *HSD17B1* gene (NC_056064.1).

A fragment containing a 5′upsteam region of human and ovine *HSD17B1* genes (−150/+9 and −150/+34) was amplified by genomic PCR. They were cloned into a pGL4.10[Luc2] vector (Promega Corporation). The pQCXIP expressing bovine HSD17B1 was generated by amplifying its open reading frame and cloning them into a pQCXIP vector (Takara Bio USA, Inc., San Jose, CA, USA). Other vectors were prepared as described [5,20,22,23].

### 2.6. Western Blot Analysis

Protein extraction and quantification were performed as described [5,20,22]. Transfected HEK293 cells were washed twice in ice-cold phosphate saline buffer and lysed in RIPA buffer (50 mm Tris-HCl (pH 8.0) containing 1% Nonidet P-40, 0.1% sodium dodecyl sulfate, 0.5% sodium deoxycholate) supplemented with 1 mm PMSF. The cell debris was removed by centrifugation at 15,000× *g* at 4 °C for 5 min, and supernatants were used as cell lysates. Proteins were quantified using a Bio-Rad Protein Assay kit (Bio-Rad Laboratories Inc., Hercules, CA, USA). Equal amounts of protein (20 μg) in each lane were separated using 10% SDS-PAGE and transferred to polyvinylidene difluoride membranes. Anti-FLAG (1:1000, M2, Sigma-Aldrich) and anti-glyceraldehyde-3-phosphate dehydrogenase (GAPDH, 1:1000; 14C10; Cell Signaling Technology Inc., Danvers, MA, USA) antibodies were used for Western blot analyses of FLAG and GAPDH proteins. We used Clarity Western ECL Substrate (Bio-Rad Laboratories Inc.) for Western blot detection on the Atto LuminoGraph II (Atto Corporation, Tokyo, Japan).

### 2.7. Statistical Analyses

All values are represented as the mean ± SEM. The one-way ANOVA followed by Tukey’s post hoc test was used for assessing the difference between the groups utilizing EZR (Saitama Medical Center, Jichi Medical University, Saitama, Japan), which is a graphical user interface for R (The R Foundation for Statistical Computing, Vienna, Austria) [21]. Individual means within treatment groups were compared using Tukey’s test. *p*-values less than 0.05 were considered significant.

## 3. Results

### 3.1. Comparison of the Placental Expression of Steroidogenic Genes in Various Mammals

We compared the expression of steroidogenic genes involved in sex steroid production in the ovaries and placentae of various mammalian species (Figure 2). Ovaries expressed the series of sex steroid-producing genes, such as CYP11A1/Cyp11a1, HSD3B1/Hsd3b, CYP17A1/Cyp17a1, CYP19A/Cyp19a1 and HSD17B1/Hsd17b1, across all species. In contrast, the placental expression of these genes was different among animal species. CYP11A1/Cyp11a1 and HSD3B/Hsd3b were expressed in all species, despite some species expressing placental HSD3B/Hsd3b isoforms (human HSD3B1 and murine Hsd3b6). However, Cyp19a1 and Hsd17b1 were not expressed in the murine placenta. Although CYP17A1 and CYP19A were expressed, HSD17B1 was almost undetectable in guinea pig, porcine, ovine, and bovine placentae. The human placenta expressed CYP19A1 and HSD17B1, whereas CYP17A1 was undetectable. These results indicate that the expression of placental HSD17B1 accounts for the differences in major circulating estrogens in pregnancy across various animal species.

### 3.2. Analyses of Human and Ovine HSD17B1 Promoter Region

In humans, SP1 and its binding sites are important for the transcriptional regulation of placental steroidogenic genes [24,25,26]. An SP1 site also occurs near the transcription start site (TSS) of human *HSD17B1* and plays a role in its transcriptional regulation (Figure 3A). In fact, promoter activities of human *HSD17B1* were increased over 40-fold compared with pGL4.10[luc2] empty vector when a reporter plasmid containing the surrounding region of this site (−150 to +9 bp from TSS) was transfected into human placenta-derived JEG-3 cells (Figure 3B). Sequence homology showed that this SP1 site is conserved in other animal species, such as mouse, porcine, bovine, and ovine (except for guinea pig, in which reliable HSD17B1 promoter sequence is unavailable). A reporter plasmid containing this site in ovine *HSD17B1* (−150 to +34 bp from TSS) was also activated in JEG-3 cells. These results indicate that in non-human species, promoter activities of *HSD17B1* are repressed by epigenetic regulation. Consistent with this hypothesis, cytosine residues of this SP1 site in the ovine *HSD17B1* promoter region were completely methylated in the placenta (Figure 3C).

### 3.3. Comparison of Enzymatic Activities of HSD17B1/Hsd17b1 for Converting A4 to T

Next, the enzymatic activities of HSD17B1/Hsd17b1 in producing T from A4 across species were compared using our established system that quantifies conversion from A4 to T on the basis of androgen receptor (AR)-mediated transactivation [5] (Figure 4). HEK293 cells were transfected with GFP or HSD17B1/Hsd17b1 expression vectors from various species (Figure 4A). At 2 days post-transfection, cells were incubated with media containing A4 at 10^−9^ M for 2 h. Then, culture media supernatants were collected and added to CV-1 cells transfected with androgen response element (ARE)-Luc and human AR expression vectors. In this system, AR-mediated luciferase activities reflect the conversion of A4 into T by genes expressed in HEK293 cells [5].

Consistent with previous studies [3], murine Hsd17b1 showed the highest enzymatic activity across species (Figure 4B). Ovine and bovine HSD17B1 also showed high activity, although significantly lower than that of murine homolog. Although supernatants of human and porcine HSD17B1-expressing cells were the lowest, they still had markedly higher activity when compared with those of GFP-expressing cells. These results indicate that mammalian HSD17B1/Hsd17b1 possess high activities for the conversion of A4 into T, albeit at different levels among various species.

## 4. Discussion

HSD17B1/Hsd17b1 is a steroidogenic gene that efficiently catalyzes the conversion of estrone (E1) into estradiol (E2). It is expressed in the ovaries of all eutherian species, whereas placental expression is significant only in humans. This expression pattern of HSD17B1/Hsd17b1 likely accounts for the fact that estrone and its sulfate form represent the major circulating estrogens in almost species, whereas estradiol is abundant in pregnant women.

Some previous studies have reported that trophoblast-like cells established from porcine fibroblasts and bovine embryos express HSD17B1 [27,28,29]. Our results suggest the possibility that this expression may be an artifact in established trophoblastic cells. Porcine fibroblast cells are partially reprogrammed to trophoblast-like cells by the introduction of Yamanaka factors (octamer-binding transcription factor 4 (OCT-4), SRY-box transcription factor 2 (SOX2), Krüppel-like transcription factor 4 (KLF4) and c-MYC), even though almost parts reprogrammed to induced pluripotent stem cells [27]. Trophoblast-like cells express various trophoblast cell lineage markers, such as caudal-type *homeobox* transcription factor 2 (CDX2), transcription factor AP-2 (TFAP2), and TEA domain transcription factor 4 (TEAD4). In addition, they also express steroidogenesis-related genes, such as CYP11A1, HSD17B1, and steroidogenic acute regulatory protein (StAR). Among them, mammalian StAR is expressed in the gonads and adrenal but not in the placenta. Therefore, porcine fibroblast-derived trophoblast cells may incompletely reflect the expression profiles of steroidogenesis-related genes in the placenta. This might also be applicable to the trophoblast stem-like cells established from bovine blastocyst using mitogen-activated kinase kinase (MEK) and glycogen synthase kinase 3 (GSK-3) inhibitors [29]. In support of this, it was reported in previous studies that E1 and its sulfate form are the predominant estrogens produced by bovine placental cells and circulated in pregnant bovine blood [12,30]. However, the possibility remains that the expression of HSD17B1 may fluctuate during the differentiation of trophoblast cells; thus, further analyses are necessary to reveal the expression of HSD17B1 and other steroidogenesis-related genes in the placenta.

A conserved SP1 site is likely the most important for the promoter activity of HSD17B1/Hsd17b1 exceeding species. However, because SP1 is ubiquitously expressed in all tissues [31], another transcription factor(s) is essential for tissue- and cell-specific expression (ovarian granulosa cells and placenta) of HSD17b1/Hsd17b1. In addition to SP1, it was reported in previous studies that various transcription factors are also involved in the transcriptional regulation of HSD17B1/Hsd17b1. In human placental cells, GATA proteins (GATA-2 and GATA-3)- and TFAP2A-binding sites repress SP1-mediated promoter activity, although their physiological significance is not clear [26]. In porcine granulosa cells, forkhead box protein A2 (FOXA2) negatively regulates the promoter activities of HSD17B1, whereas p53 positively regulates that [32]. Because these transcription factors are expressed in various tissues and cells other than ovarian granulosa cells, their contribution to granulosa cell-specific expression remained unclear. It is interesting that estrogen-related receptor γ (ESRRG) was reported to regulate the transcription of *HSD17B1* [33]. ESRRG is highly expressed in the human placenta, and its reduced expression is possible to be involved in fetal growth restriction and preeclampsia [34,35]. ESRRG can induce HSD17B1 expression by direct binding to its promoter region in the human placental cell line, HTR-8/SVneo cells. However, this phenomenon is also unlikely the cause of human placenta-specific HSD17B1 expression. ESRRG/Esrrg is expressed in the placenta of other species [34]. In addition, there are potentially multiple ESRRG/Esrrg-binding sites in the proximal 5´upsteam region of the *HSD17B1*/*Hsd17b1* gene of other mammalian species, such as mouse, porcine, bovine, and ovine (our unpublished data). Further studies are necessary to reveal the mechanisms of ovarian and placental HSD17B1 expression in the future. In contrast, it is conceivable that epigenetic regulation via the methylation of cytosines at the CpG dinucleotides in the promoter region is very important in tissue- and species-specific HSD17B1/Hsd17b1 expression. In addition to the ovary and placenta, human HSD17B1 is highly expressed in some tumors and involved in the E2-mediated tumor progression [36]. In non-small cell lung cancer (NSCLC), expression of HSD17B1 mRNA and proteins is markedly increased, causing tumor cell proliferation through enhanced E2 production [37]. Upregulation of HSD17B1 expression in NSCLC cells is induced by DNA demethylation of the *HSD17B1* promoter using 5-Aza-2′-deoxycytidine. In contrast, HSD17B1 expression is decreased in colorectal cancer through methylation of the promoter region, resulting in tumor progression via E2 reduction [38]. Taken together, it is probable that expression of HSD17B1/Hsd17b1 is basically repressed in many tissues and cells by the methylation of cytosines at the CpG dinucleotides in the proximal 5’upsteam region.

Human HSD17B1 can convert A4 into T, similar to its homologs in other mammalian species, although to a lesser extent than murine Hsd17b1, which possesses strong activity that is comparative efficiency for converting E1 to E2. This is inconsistent with previous studies, which indicate that the catalytic efficiency of human HSD17B1 for the conversion of A4 to T is very weak or barely detectable (about one-fourth lower *V_max_* and 27-fold higher *K*_m_ than rodent homolog), although it has a similar or stronger activity for converting E1 to E2 compared with rodent homolog [3,4,39]. However, since such results were derived using A4 at supraphysiological high substrate concentrations (100–700 nM), it is conceivable that human HSD17B1 also has a relatively high activity for converting A4 into T under physiological conditions, as demonstrated in this study (A4 at 1 nM). In support of this hypothesis, human HSD17B1-transgenic female mice show increased T concentrations preceding birth with some features of masculinization [40]. Furthermore, the phenotypes of placental aromatase-deficient patients suggest that rather than E2 production, T production is physiologically significant for placental HSD17B1. Human placental aromatase deficiency (mutations of the fetal *CYP19A1* gene) causes virilization of the mother and female fetus in utero as a result of androgen excess secondary to impaired conversion to estrogens [41]. However, mothers deliver infants at term in almost all cases [8], even though androgen excess during pregnancy subsequently impairs postnatal growth and development [42]. Based on these results, it is proposed that placental aromatase protects the pregnant mother and fetus from androgen excess during pregnancy. This hypothesis also suggests that placenta-derived estrogens are unessential for pregnancy and delivery. Therefore, it is conceivable that the most important function of human placental HSD17B1 is likely the production of T rather than E2. In fact, plasma T levels in conjugation with other androgens progressively increase during pregnancy [43]. These androgens play important roles in fetal development. In addition, they are also involved in the myometrial contraction and cervical remodeling in the fetus during parturition [43]. However, further studies are necessary to reveal the importance of androgens during pregnancy. 

Compared with its physiological significance, the deleterious impact of T and other androgens during pregnancy has been rigorously outlined in various animal species. T excess causes virilization, hypertension, and metabolic and reproductive dysfunction not only in mothers [44] but also in the fetus [45,46]. In addition, the impacts of prenatal T exposure are continued during postnatal life, leading to abnormal reproductive cycles with metabolic syndrome in females [46]. Polycystic ovary syndrome (PCOS) is a common disease in women of reproductive age with such symptoms [45,47]. The ovaries in PCOS women accumulate many small follicles by abnormal folliculogenesis, often resulting in oligo-anovulatory infertility. Although the exact cause of PCOS remains unknown, many patients have excessive androgens derived from the ovaries and adrenal. Consistent with this fact, animal models for PCOS are induced by the administration of T and other androgens in various species [48]. It was recently reported that the offspring of PCOS patients and female mice with PCOS-like characteristics induced by androgen treatment during pregnancy are susceptible to PCOS or PCOS-like traits with obesity [49]. In the murine model, this phenotype is transmitted to the second and third generations from androgen-exposed mothers. Based on these facts, it is conceivable that excess T during pregnancy is harmful to the survival of eutherian species. Deficiency of CYP17A1 and HSD17B1 expression should be vital to preventing de novo T synthesis in the placenta of humans and other animal species, resulting in eliminating the risk of T excess. Again, this hypothesis would be supported by the phenotypes of human *HSD17B1*-transgenic female mice [40]. In the future, it may be worth investigating why placentally repressed genes for the prevention of T production were different between humans and other animal species. It is clear that epigenetic regulation is important for the selective repression of placental steroidogenic genes. As in the case of *HSD17B1* in the ovine placenta, the promoter region of *CYP17A1* is methylated in human placenta cells, resulting in the repression of CYP17A1 expression [50]. Although it was reported that CYP17A1 is detectable in the human placenta [51], various previous studies [8,50] and our results indicate that expression levels of human placental CYP17A1 are too low to contribute to de novo androgen synthesis by the placenta during pregnancy. In contrast to the regulation of gene expression, the physiological significance of this phenomenon is unknown. The placenta is important for pregnancy in all eutherian species, whereas their cellular constituents and characteristics are markedly different between species [6,52,53]. It is essential to elucidate the relationship between placental physiology and steroid hormone production using additional animal species in future studies.

## 5. Conclusions

HSD17B1 is expressed in the human placenta, whereas it was almost undetectable in the placenta of other eutherian species. DNA methylation analyses indicate that epigenetic regulation is important for this selective placental gene expression. Human HSD17B1 has high activity for the conversion of A4 into T, similar to its homologs in other species. Since placenta-derived estrogens are not essential for pregnancy and delivery, our current results provide novel insights into the role of placental HSD17B1 in active androgen production.

## Figures and Tables

**Figure 1 animals-13-00622-f001:**
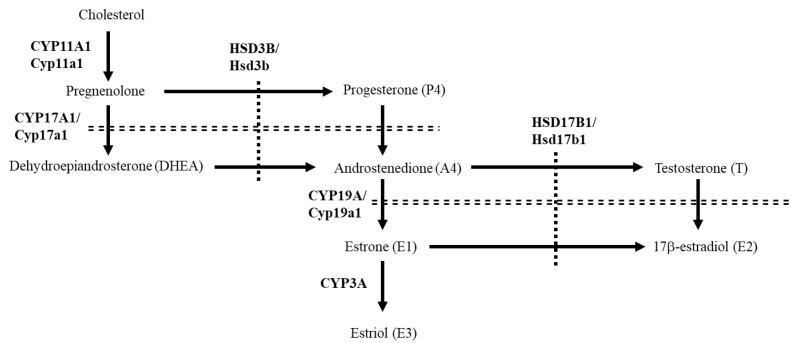
Pathways for producing androgens and estrogens in mammalian ovary and placenta. Human, bovine, ovine, and porcine steroidogenic enzymes are indicated totally by capital letters, whereas rodent counterparts are indicated only initially by a capital letter.

**Figure 2 animals-13-00622-f002:**
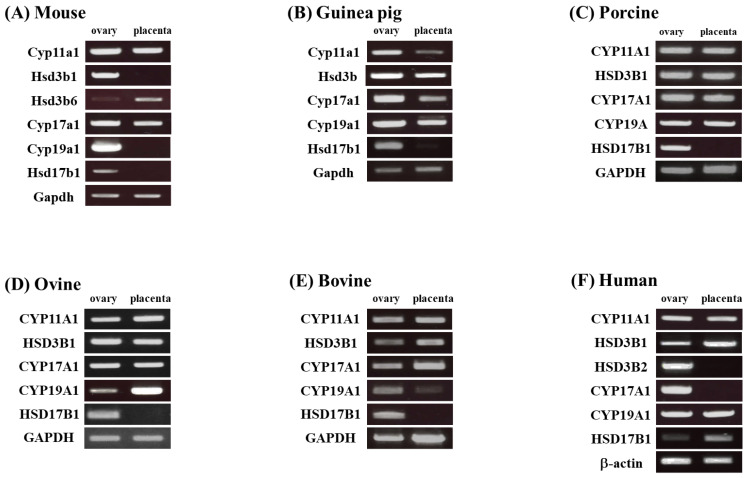
Expression of steroidogenic genes in ovary and placenta of various mammalian species. RT-PCR analyses of each gene in the ovary and placenta of a mouse (**A**), guinea pig (**B**), porcine (**C**), ovine (**D**), bovine (**E**), and human (**F**).

**Figure 3 animals-13-00622-f003:**
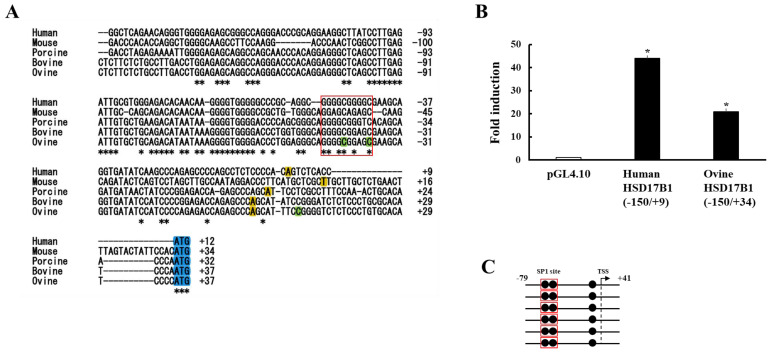
Analyses of HSD17B1 promoter region (**A**) Alignment for the nucleotide sequence of HSD17B1 gene and its 5′-flanking region in human, mouse, porcine, bovine, and ovine. The asterisks (*) are the conserved nucleotides among all species. The conserved SP1 site is shown by a redline box. Putative transcription start and translation start sites are shown by the yellow and blue boxes, respectively. Cytosine bases in CpG dinucleotides of ovine HSD17B1 promoter region are shown by green boxes. (**B**) Activation of human and ovine HSD17B1 promoter in human placenta-derived JEG-3 cells. Each vector was transfected by lipofection into JEG-3 cells. At 48 h after transfection, luciferase assays were performed using the cell lysates. Values of the pGL4.10 Basic vector were defined as 1. Data are the mean  ±  SEM values of four independent experiments * *p* < 0.05 vs. pGL4.10. (**C**) Methylation analysis of the promoter region (−79 to +41) of the HSD17B1 gene in the ovine placenta. The methylation pattern of individual bisulfite-sequenced clones is indicated. Each *circle* denotes cytosine bases in CpG dinucleotides, and *filled circles* represent methylated and unmethylated cytosines, respectively.

**Figure 4 animals-13-00622-f004:**
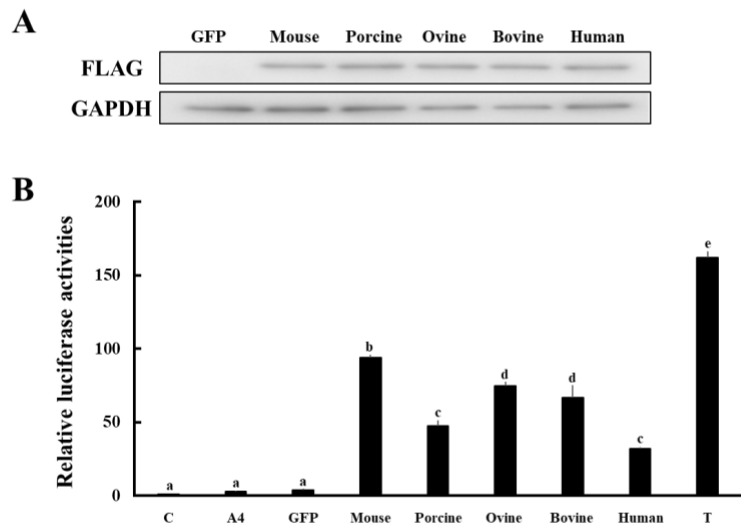
Comparison of enzymatic activities of HSD17B1/Hsd17b1 in various species. (**A**) Western blot analyses were performed with the antibodies against FLAG-tag and GAPDH using lysates of GFP- and FLAG-tagged HSD17B1/Hsd17b1 (mouse, porcine, ovine, bovine, and human)-introduced HEK293 cells. (**B**) Activation of AR-mediated transcription by culture media from GFP or each HSD17B1/Hsd17b1-introduced HEK293 cells collected at 2 h after A4 addition. CV-1 cells were transfected with ARE-Luc and AR expression vectors. At 24 h post-transfection, cells were incubated with vehicle (lane C), A4 (1 nM), T (1 nM), or culture medium from GFP- and each species HSD17B1/Hsd17b1-expressing HEK293 cells for 24 h. Values of the vehicle were defined as 1. Data represent the mean ± SEM of four independent experiments. Values marked by the different letters are significantly different from each other (*p* < 0.05).

## Data Availability

The data presented in this study are available within the article.

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
