# Peer review of "Comparison of Placental HSD17B1 Expression and Its Regulation in Various Mammalian Species"

_animals, 2023, doi:10.3390/ani13040622_

Round 1

Reviewer 1 Report

The present work shows an experiment comparing the expression of genes from different species, several issues must be addressed for a possible further re-evaluation:

On methods it is not clear in which stage of development placenta samples was taken, for example, the endocrinological function of ovine placenta changes accordingly with the stage of pregnancy, also the developmental stage of the placenta differs between species, also placenta of pigs and ruminants differs from human and rodents, which stage in days was collected and what part of the placenta was used? 

Bovine primers might be not correctly reported, for example using the primer blast tool based on the supplementary file the primers for bovine HSD3B1 returns HERC1, and bovine HSD17B1 returns bovine HSD3B1. 

Discussion:

The paragraph from lines 263 to 278 is based on the premise that bovine HSD17B1 is not expressed in the placenta. The authors fail to address this issue because, when looking to the embryogene profiler it is possible to see the increased expression of bovine HSD17B1 in whole blastocysts, which are constituted in at least 2/3 of trophectoderm cells. Furthermore, on data from " Moradi M, Zhandi M, Sharafi M, Akbari A et al. Gene expression profile of placentomes and clinical parameters in the cows with retained placenta. BMC Genomics 2022 Nov 21;23(1):760 " it is possible to see the bovine HSD17B1 relatively highly expressed in bovine placenta. 

Concluding the work in the present format is not suitable for publication once it is not clear how samples were collected, there is not much information about the samples, discussion and conclusion were not written in-depth enough, for example, "HSD17B1 is expressed in human placenta, whereas it was almost undetectable in placenta of other eutherian species.", the author's conclusion is not backed by the literature and the present data fail to prove the contrary using only RT-PCR. Moreover, the methodology of RT-PCR and Primers information must be properly addressed. 

Author Response

We thank Reviewer1 for the valuable comments. We revised the manuscript as follows.

â‘ On methods it is not clear in which stage of development placenta samples was taken, for example, the endocrinological function of ovine placenta changes accordingly with the stage of pregnancy, also the developmental stage of the placenta differs between species, also placenta of pigs and ruminants differs from human and rodents, which stage in days was collected and what part of the placenta was used? 

   →As suggested, we added the information of used placenta to Materials and Methods section. We used bovine ovine and porcine placenta just after parturition, whereas rodent placenta were obtained during middle to later pregnancy (13-15 and 20-25 dpc in mice and guinea pigs, respectively). Because human placenta RNA was purchased from supplier, it is impossible to get the detail information, except for age of pregnant mothers (23-30 years old). As the reviewer said, placental development is different between species. However, they can produce progesterone and estrogens at these stages, despite the amounts should be fluctuated. Basically, we used total RNA extracted from whole ovary and placenta homogenized in RNA extraction reagents.

â‘¡Bovine primers might be not correctly reported, for example using the primer blast tool based on the supplementary file the primers for bovine HSD3B1 returns HERC1, and bovine HSD17B1 returns bovine HSD3B1. 

      →We are sorry for our mistakes. We corrected the pointed primer sequences (Supplementary Table 1).

â‘¢Discussion:                                                                 The paragraph from lines 263 to 278 is based on the premise that bovine HSD17B1 is not expressed in the placenta. The authors fail to address this issue because, when looking to the embryogene profiler it is possible to see the increased expression of bovine HSD17B1 in whole blastocysts, which are constituted in at least 2/3 of trophectoderm cells. Furthermore, on data from " Moradi M, Zhandi M, Sharafi M, Akbari A et al. Gene expression profile of placentomes and clinical parameters in the cows with retained placenta. BMC Genomics 2022 Nov 21;23(1):760 " it is possible to see the bovine HSD17B1 relatively highly expressed in bovine placenta. 

      →It is difficult to evaluate the expression of HSD17B1 in EmbryoGENE Profiler, because about half of the blastcyst samples are within the range of background level. Although we have checked the concerned paper, there are no description and data of placental HSD17B1 expression. Rather, it is difficult to explain the previous reports that E1 and its sulfate form are the predominant estrogens produced by placental cells (Matins-Junior et al., Endocrinology, 2014; Matamoros et al., Bio Reprod,1994) and circulated in pregnant cow blood (Martins-Júnior et al., Endocrinology, 2014), if HSD17B1 is highly expressed in the bovine placenta. These facts could support our conclusion. We added the discussion about these previous studies in Discussion section (P8-L286-289).

â‘£Concluding the work in the present format is not suitable for publication once it is not clear how samples were collected, there is not much information about the samples, discussion and conclusion were not written in-depth enough, for example, "HSD17B1 is expressed in human placenta, whereas it was almost undetectable in placenta of other eutherian species.", the author's conclusion is not backed by the literature and the present data fail to prove the contrary using only RT-PCR. Moreover, the methodology of RT-PCR and Primers information must be properly addressed. 

   ã€€ã€€→We agree that our results do not completely reject the possibility of HSD17B1 expression in the eutherian species except human. Therefore, we described in the Discussion section that ‘further analyses are necessary to reveal expression of HSD17B1 and other steroidogenesis-related genes in the placenta`(P8-L290~291). However, previous studies never perform even RT-PCR analyses to investigate the expression of placental HSD17B1 in these species. We can’t detect HSD17B1 expression in multiple placenta of these species, even though CYP11A1 and other steroidogenic genes were expressed. In addition, as we mentioned above, the results of many previous studies are consistent with our data. Hence, we think this study includes some significant insights into the placental steroidogenesis.

Reviewer 2 Report

MDPI Animals: Yazawa T. et al. ‘Comparison of placental HSD17B1 expression and its regulation in various mammalian species’.

This manuscript reports a study using RT-PCR to examine the level of expression of various enzymes of steroidogenesis in ovaries and placentas of six species of mammals, including humans. This is an interesting new approach in comparative endocrinology which is complementary to identification/quantification of circulating steroids in the plasma. The findings of the study provide a possible biochemical explanation as to why the placentas of many animals produce oestrone, while human placentas produce oestradiol. However, the manuscript lacks some details and may be somewhat preliminary.

Section 2.1: Where and how were the human samples sourced? How many samples of each animal species were used? How many replicates of each? To what stage of pregnancy do each of the samples correspond, as expression of the various genes may vary during the course of pregnancy?

Section 2.3: The two-stage assay for conversion of A4 to T needs to be explained more clearly as it took me several readings to work out how it was functioning.

Section 3.2: Findings on the mouse and guinea pig HSD17B1 promoters are missing from this section. They should be included to maintain the full comparative aspect of the study. Why are there 6 identical lines in Fig. 3c for methylation analysis of ovine HSD17B1 promoter region?

Section 3.3: The study would be beneficially extended and improved by inclusion of a parallel assay of the conversion of E1 to E2 by using the first stage of the assay for T with either an ER-based second assay or quantification of produced E2 by HPLC-MS. This would establish whether the HSD17B1 enzymatic activities from each species have the same relative activities with A4 and E1; the statement that ‘AR-mediated luciferase activities reflect the conversion of A4 to T’ needs to be substantiated by citation of appropriate reference(s).

Section 4: barely detectable (line 283 & line 60); the proposal that the function of human placental HSD17B1 may be to produce T needs to be developed and substantiated further; How is this compatible with the later statement (line 321) that ‘expression levels of human placental CYP17A are too low to contribute to androgen production during pregnancy’? ; lines 283-287: the enzyme has a low Vmax/turn-over rate?; ‘cervical remodelling in the foetus’?

References: The initial letters of all the major words in journal titles need to be capitalised.

Figure S1: Explain ‘O’ and ‘P’ in the legend

English: The English of the manuscript is somewhat stilted with the result that some sentences are difficult to follow or do not make sense. The authors should obtain the help of a native English-speaking scientist to improve it.

Author Response

We thank Reviewer2 for the valuable comments. We revised the manuscript as follows.

Section 2.1: Where and how were the human samples sourced? How many samples of each animal species were used? How many replicates of each? To what stage of pregnancy do each of the samples correspond, as expression of the various genes may vary during the course of pregnancy?

        →Total RNA from human tissues were purchased from Takara Bio Inc and BioChain Institute Inc. (P3-L124~126). As suggested, we added the sample number of each species (n=2~8). We performed at least triplicate experiments. We used human, bovine ovine and porcine placenta just after parturition, whereas rodent placenta were obtained during middle to later pregnancy (13-17 and 20-25 dpc in mice and guinea pigs, respectively). We added this information (P3-L113~127, L131~132). As the reviewer said, gene expression pattern is changed during pregnancy. However, it is conceivable that the expression of steroidogenic genes is maintained, because they can produce progesterone and estrogens at these stages.

Section 2.3: The two-stage assay for conversion of A4 to T needs to be explained more clearly as it took me several readings to work out how it was functioning.

         →As suggested, we added the detail explanation of the cell-based assays as follows (P4-L141~148);

     In the experiments for evaluating the enzymatic activies of HSD17B1/Hsd17b1, we adopted a method that quantifies conversion of A4 into T via AR-mediated transactivation [5]. GFP or HSD17B1/Hsd17b1 expression vectors were transfected into HEK293 cells. After 2 days post-transfection, HEK293 cells were treated with A4 at 10-9 M for 2 h. CV-1 cells were transfected with ARE-Luc and human AR- expression vectors. After 24 h post-transfection, CV-1 cells were incubated for 24 h with vehicle (EtOH), A4 (1 nM), T (1 nM), culture medium derived-from GFP or each HSD17B1/Hsd17b1-expressing HEK293 cells collected at 2 h after A4 addition.

Section 3.2: Findings on the mouse and guinea pig HSD17B1 promoters are missing from this section. They should be included to maintain the full comparative aspect of the study. Why are there 6 identical lines in Fig. 3c for methylation analysis of ovine HSD17B1 promoter region?

→As suggested, we added the mouse Hsd17b1 promoter in Fig. 3A. Unfortunately, reliable sequence of guinea pig HSD17B1 promoter is not unrevealed still now.

   Fig. 3C indicated the methylation pattern of pattern of 6 individual bisulfite-sequenced clones. We added this point to the Figure legends (P6-L228).

Section 3.3: The study would be beneficially extended and improved by inclusion of a parallel assay of the conversion of E1 to E2 by using the first stage of the assay for T with either an ER-based second assay or quantification of produced E2 by HPLC-MS. This would establish whether the HSD17B1 enzymatic activities from each species have the same relative activities with A4 and E1; the statement that ‘AR-mediated luciferase activities reflect the conversion of A4 to T’ needs to be substantiated by citation of appropriate reference(s)

     →We agree it is interesting to compare the HSD17B1 activities of various species to produce E2. Unfortunately, we have never established the ER-mediated system to evaluate the enzymatic activities for converting E1 to E2, yet. In addition, time is not enough to measure E2 using HPLC-MS. However, it was reported in multiple studies that human HSD17B1 has than rodent homologs. We added this fact to the Discussion section as follows (P8-L294-298);

      This is inconsistent with previous studies, which indicate that the catalytic efficiency of human HSD17B1 for the conversion of A4 to T is very weak or barely detectable (about one-fourth lower Vmax and 27-fold higher Km than rodent homolog), although it has similar or stronger activity for converting E1 to E2 compared with rodent homolog [3,4,31].

     →We cited our previous work in P7-L240, [5] (Yazawa et al. JSBMB, 2020).

Section 4: barely detectable (line 283 & line 60); the proposal that the function of human placental HSD17B1 may be to produce T needs to be developed and substantiated further; How is this compatible with the later statement (line 321) that ‘expression levels of human placental CYP17A are too low to contribute to androgen production during pregnancy’? ; lines 283-287: the enzyme has a low Vmax/turn-over rate?; ‘cervical remodelling in the foetus’?

         →Thank you for valuable comments. Human placental CYP17A1 are too low to contribute to de novo androgen synthesis by the placenta during pregnancy. We re-write this point (P8-L335-336).

the enzyme has a low Vmax/turn-over rate?

→As suggested, human HSD17B1 has a low Vmax and high Km for converting A4 to T, compared with rodent homolog. We also added this point as follows (P8-L296~297);

(about one-fourth lower Vmax and 27-fold higher Km than rodent homolog)

‘cervical remodelling in the foetus’?

         →Thank you for a valuable comment. We corrected this point (P8-L317).

References: The initial letters of all the major words in journal titles need to be capitalised.

          →As suggested, we corrected this point.

Figure S1: Explain ‘O’ and ‘P’ in the legend

        →As suggested, we added this point to Figure S1.

English: The English of the manuscript is somewhat stilted with the result that some sentences are difficult to follow or do not make sense. The authors should obtain the help of a native English-speaking scientist to improve it.

         →As suggested, our manuscript was edited by native speakers of Edanz.

Reviewer 3 Report

In this work, the authors compared the expression of HSD17B1 in various mammalian species.

The authors should include the age of the animals used for the study in the "materials and methods" section.

They should also specify the dilution of antibodies used for WB and the amount of cDNA used for PCR.

Author Response

We thank Reviewer3 for the valuable comments. We revised the manuscript as follows.

The authors should include the age of the animals used for the study in the "materials and methods" section.

          →As suggested, we added the age of the animals in the Materials and Methods section.

They should also specify the dilution of antibodies used for WB and the amount of cDNA used for PCR.

            →As suggested, we added these points (P3-L131-132, P4-L173~174). We used 1:1000 diluted antibodies in WB analyses. In PCR amplification, we used 25 ng cDNA/ reaction as a template.

Round 2

Reviewer 1 Report

Yazawa et.al. addressed my questions well, and they discussed the issues based on the literature. Although some questions will remain uncertain they can be addressed in future experiments.

Author Response

We thank Reviewer1 for the positive comments. We would like to try to reveal the problems in the future study.

Reviewer 2 Report

MDPI Animals: Yazawa T. et al. ‘Comparison of placental HSD17B1 expression and its regulation in various mammalian species’ (Resubmission).

Only a few minor corrections/additions are now needed:

L45: ‘epigenetic regulation’ is written twice.

L48: change ‘discussed’ to ‘discuss’.

L79: represent the main source.

M&M; state that experiments were done at least in triplicate.

L136: previously.

L229: The methylation pattern of 6 individual bisulfite-sequenced clones are indicated.

Figure 3 or text: Explain briefly the lack of guinea pig data in Panel A.

L288: reported.

Author Response

We thank Reviewer2 for the valuable comments. We revised the manuscript as follows. We showed corrected parts of the text in red letters.

L45: ‘epigenetic regulation’ is written twice.

   →Thank you for valuable comments. As suggested, we corrected this point (P1-L44~45).

L48: change ‘discussed’ to ‘discuss’.

    →As suggested, we changed this point (P1-L47).

L79: represent the main source.

    →As suggested, we changed this point (P2-L78).

M&M; state that experiments were done at least in triplicate.

    →As suggested, we added this point (P3-L129~130).

L136: previously.

    →We corrected this point (P3-L136).

L229: The methylation pattern of 6 individual bisulfite-sequenced clones are indicated.

    →As suggested, we changed this point (P6-L230).

Figure 3 or text: Explain briefly the lack of guinea pig data in Panel A.

      →As suggested, we added this point in text (P6-L213~215).

L288: reported.

     →We corrected this point (P8-L288).